# Feasibility Study of a Menstrual Hygiene Management Intervention for People with Intellectual Impairments and Their Carers in Nepal

**DOI:** 10.3390/ijerph16193750

**Published:** 2019-10-04

**Authors:** Jane Wilbur, Thérèse Mahon, Belen Torondel, Shaffa Hameed, Hannah Kuper

**Affiliations:** 1International Centre for Evidence in Disability, London School of Hygiene & Tropical Medicine, London, WC1E 7HT, UK; shaffa.hameed@lshtm.ac.uk (S.H.); Hannah.Kuper@lshtm.ac.uk (H.K.); 2WaterAid, 47-49 Durham Street, London, SE11 5JD, UK; Theresemahon@wateraid.org; 3Environmental Health Group, Department of Clinical Research, London School of Hygiene & Tropical Medicine, London, WC1E 7HT, UK; Belen.Torondel@lshtm.ac.uk

**Keywords:** menstrual hygiene management, intellectual impairment, carers, adolescent, young people, disability, behaviour change, feasibility study, behaviour-centred design

## Abstract

*Background:* The Bishesta campaign is a menstrual hygiene management (MHM) intervention developed to meet the specific needs of people with intellectual impairments and their carers. It was designed and delivered in the Kavre district, Nepal. This paper explores the campaign’s feasibility and acceptability. *Methods:* The Bishesta campaign was delivered to ten people with an intellectual impairment and their eight carers. Data on the feasibility and acceptability of the intervention was collected through: Structured questionnaire to participants before and after the intervention, process monitoring data, post-intervention in-depth interviews with all carers, observation of people with intellectual impairments, key informant interviews with all facilitators and staff involved in the campaign, as well as ranking of the perceived appropriateness and acceptability of campaign components by carers and facilitators. *Results:* The Bishesta campaign was acceptable for the target groups, facilitators, and implementers. It was largely delivered with fidelity. Participants used most of the campaign components; these made the target behaviours attractive and enabled participants to carry them out with ease. There were improvements across all target behaviours. The focus of this study was feasibility, not limited-efficancy; however, indicative positive outcomes from this small sample were observed, such as an increase in young people’s levels of confidence, comfort, and autonomy during menstruation. *Conclusion:* Within the sample, the Bishesta campaign appears to be a feasible intervention to ensure that one of the groups most vulnerable to exclusion from MHM interventions is not left behind.

## 1. Introduction

### 1.1. Background

An estimated one billion people live with a disability worldwide [1]. In low- and middle-income countries, many people with disabilities have inadequate access to water, sanitation, and hygiene (WASH) services [2,3,4,5]. WASH services are vital for effective menstrual hygiene management (MHM). MHM is defined as women and adolescent girls using a clean menstrual management material to absorb or collect blood that can be changed in privacy as often as necessary for the duration of the menstruation period, using soap and water for washing the body as required, and having access to facilities to dispose of used menstrual management materials. They understand the basic facts linked to the menstrual cycle and how to manage it with dignity and without discomfort or fear” [6]. MHM also involves addressing associated harmful social beliefs and taboos.

Menstruation is a sign of good health, but it is shrouded in taboo and secrecy [7,8,9,10]. In many countries, menstrual blood is considered polluting [10] and people with disabilities are also considered ‘dirty’ and ‘contagious’ [11,12], and so may face double taboos when menstruating. At the onset of menarche, girls are taught ‘menstrual etiquette’, which encourages secrecy and discomfort, as well as limited formal puberty guidance [9], and perpetuates difficulties in MHM.

### 1.2. Disabling Menstrual Hygiene Barriers Research

People with disabilities may not be included in training on menstrual etiquette, whether through preconceptions that they will not menstruate, exclusion from school, or difficulties understanding. They may also face additional difficulties in managing their menstruation independently, as a result of physical, sensory, or intellectual impairments. Consequently, people with disabilities may have additional needs for MHM, but there has been a lack of research investigating this issue in low- and middle-income countries (LMICs).

The disabling menstrual hygiene barriers research aims to investigate and address the barriers to MHM that people with disabilities face in Nepal. Drawing on the behaviour change design model in its design [13], research activities conducted include: (1) Completing a systematic review of relevant peer reviewed literature on the MHM requirements of people with disabilities in different settings, and the barriers that they face [14]; (2) formative research to understand the specific MHM requirements of (a) adolescents and young people with a disability, and the barriers they face in managing their menstruation hygienically and with dignity in the Kavre district, Nepal, and (b) carers who support these people during menstruation [15]; (3) identifying strategies to improve MHM of people with intellectual impairments in Nepal; (4) developing the Bishesta campaign—an MHM behaviour change intervention that aims to enable people with intellectual impairments to manage their menstruation more independently, and carers to support another person’s menstrual cycle [16]; (5) delivering the MHM behaviour change intervention in Nepal and collecting process monitoring data over a three-month timeframe; and (6) conducting a process evaluation and a feasibility study to assess the feasibility and acceptability of the intervention.

Key findings from the systematic review and the formative research revealed: People with intellectual impairments face particular barriers to MHM; limited MHM interventions for this group exist; carers have limited understanding of severity of pre-menstrual symptoms experienced by people with intellectual impairments, and that they receive no information or guidance about how to support the management of someone else’s menstrual cycle. Consequently, an MHM behaviour change intervention targeting people with intellectual impairments and their carers was developed by professionals experienced in developing and delivering WASH, MHM, and/or disability interventions and marketing in Nepal. This included the Down Syndrome Society Nepal (DSSN) and the Centre for Integrated Urban Development (CIUD), who also delivered the intervention. For more details on how the intervention was developed, see Wilbur and Bright [16].

#### The Intervention and Its Delivery

The Bishesta campaign’s target group are adolescent and young people, aged 15–24 years, who menstruate, have an intellectual impairment, and live in the Kavre district and their carers. Three target behaviours were identified from the formative research for people with intellectual impairments (referred to as ‘young people’ hereafter): (1) Use a menstrual product, (2) use pain relief for menstrual discomfort, and (3) do not show menstrual blood in public. Three target behaviours were also identified for carers: (1) Provide enough clean menstrual products, (2) provide pain relief, and (3) show love and emotional support to the young person when she is menstruating. The campaign is based on two characters: Bishesta (meaning ‘extraordinary’ in Nepali), who has an intellectual impairment and menstruates, and Perana (meaning ‘motivation’ in Nepali), Bishesta’s carer. Both characters embody the target behaviours.

The intervention included ‘period packs’ for the young people. These contained: (1) A branded bag for storing clean menstrual pads at home; (2) a shoulder bag for carrying menstrual pads outside the home, with a plastic bag inside to transport used menstrual products; (3) a flip top bin, for disposing used menstrual pads at home; (4) a pain bangle with three strips of colour representing severity of menstrual discomfort—red for severe, orange for moderate, and yellow for mild; and (5) two visual stories about Bishesta menstruating and learning to manage as independently as possible, with Perana’s support. Carers received a menstrual calendar for tracking the young person’s menstrual cycle.

The campaign was delivered by five facilitators from the DSSN and the CIUD, under the guidance of the lead author and WaterAid Nepal. The campaign was implemented through group training sessions for participants and monitoring visits to the young person’s private or care home. The delivery process was monitored.

The aim of this study was to assess the feasibility of the Bishesta campaign in the Kavre district, Nepal, by investigating its acceptability, demand, implementation, and practicality [17].

## 2. Materials and Methods

### 2.1. Research Design

#### 2.1.1. Feasibility Study

Process monitoring data were collected during implementation to understand (a) if the campaign was delivered as intended, (b) how many of the people attended every session and how often they were exposed to the campaign, and (c) the extent to which participants were using the campaign components.

Bowen et al.’s feasibility study framework identifies eight areas of focus: Acceptability, demand, implementation, practicality, adaptation, integration, expansion, and limited-efficacy testing [17]. This study applies this framework but only focuses on the first four areas of focus (see Table 1); the remaining areas are outside the parameters of this research and should be explored in the future (i.e., adaptability, integration, expansion, and limited-efficancy). Within the ‘practicality’ area of focus, this study also assessed behaviour change against the target behaviours. The feasibility study indicators and a summary of the results are included in Appendix A, with results from the household monitoring indicators are captured under indicator 1.

#### 2.1.2. Study Site and Participants

Ethical approval was granted from the Nepal Health Research Council (code 39-2018) and the LSHTM Ethics Board (code 15703).

The feasibility study on the Bishesta campaign was conducted in December 2018. Participants were recruited from the formative research sample and through the DSSN’s networks. Study participants were ten young people with an intellectual impairment, aged between 15–24 who menstruated. Six young people were from a residential care home, and four were from households in the Kavre district, Nepal. The Washington Group Short Set of questions were administered with carers and used to identify young people who experience ‘a lot of difficulty’ (70%, *n* = 7) or ‘cannot do at all’ (30%, *n* = 3) across the ‘remembering and concentrating’ functional domain [18]. People who ‘cannot do at all’ are classified as having a severe intellectual impairment in this article.

Eight of the young people’s carers were recruited. These carers were professionals (50%, *n* = 4) who worked in a rural residential care home, or mothers (38%, *n* = 3) and sisters (13%, *n* = 1) living at home in urban areas with the young person.

Five facilitators who delivered the intervention and three members of WaterAid staff (Nepal and UK), with knowledge of active involvement in the intervention, were also included as research participants.

Informed consent was obtained from the carers and key informants before enrolment. Assent was sought from the young person, and consent was given by their carers. It was made clear to all participants that their involvement was voluntary.

#### 2.1.3. Data Collection Tools

During the delivery of the intervention, the facilitation team administered process monitoring data collection tools to record (1) participants’ attendance of the training sessions, (2) if the training sessions were delivered as intended (i.e., number of sessions delivered, if the correct number of facilitators delivered the sessions, if all the resources were available), and (3) household monitoring visit indicators (Table 2). If indicators 8 or 9 were achieved, the household was classified as a “Bishesta household”.

Facilitators visited the young person and their carer after each group training session to gather the household monitoring data, re-emphasise information shared at the group training sessions, answer any questions participants may have, and support them to achieve the target behaviours.

The lead author, with support from and translation by Anita Sigdel (a woman with a visual impairment involved in the formative research data collection), administered the following mixed methods data collection tools during the feasibility study:(a)Structured questionnaire exploring the target behaviours, administered before and after the intervention(b)In-depth interviews and ranking campaign components with carers. Carers were asked to rank the campaign components according to which they used most to least often, what they liked and disliked about the components, and how they could be improved. Carers then ranked the components according to which (1) the young person used least to most and why, and (2) they felt led to the biggest change in the young person’s target behaviours(c)Observation of young people. In an attempt to encourage the young people to express how they viewed each campaign component, the lead author asked the young person to select an emoji ball (happy, neutral, or sad) that most closely represented their feelings about each campaign component. However, this was not understood, so faces with the same expressions were drawn on a piece of paper and the exercise repeated. This was not understood either, so the lead author passed one campaign component at a time to the young person, and the research team observed and recorded their responses, including recognition, affinity, and understanding. For instance, if a young person smiled when passed a visual story, flicked through it, and indicated what was happening on each page, the researchers assumed that she had a level of understanding of the story(d)Key informant interviews were conducted by the lead author with all facilitators and WaterAid staff; facilitators also ranked the campaign components according to which they thought was the most useful within the group training sessions

#### 2.1.4. Data Analysis

Data from observing the young people’s reactions to the campaign components was discussed by the research team after each interaction, to reach a consensus about the findings and minimise researcher bias. After each day of fieldwork, interview notes were reviewed by the research team. This helped to identify gaps in the interview schedule and emerging themes. In-depth and key informant interviews were translated from Nepali into English and then transcribed. These transcriptions were checked by English speaking Nepali people. A thematic analytical approach was used to analyse findings. Drawing on Braun and Clarke [19], the lead author read the interview transcripts, coded data into emerging themes, and reviewed and refined them before finalising and naming the themes. Data was coded using NVivo 11 (QSR International, Warrington, UK); data were analysed to develop a fuller framework of themes and sub-themes. Relevant quotations are presented in this article. Data analysis of the quantitative data was conducted in Microsoft Excel (Microsoft, Washington, United States).

## 3. Results

### 3.1. Acceptability

Acceptability is the extent to which those delivering and receiving the intervention found it satisfying and appropriate [17].

#### 3.1.1. Satisfaction

All facilitators reported that the training they received prepared them to deliver the campaign. The majority of facilitators (80%, *n* = 4) said that the level of detail in the campaign manual was helpful as it reminded and guided them to deliver the campaign as intended; it also increased their confidence and they did not find the detail overwhelming.

Young people benefited from the practical activities and visual methods, but these also benefited the carers, some of whom had limited levels of literacy.

*“We were kind of afraid [……]. If there were only theoretical sessions, I would have felt bored and would not have understand what they were trying to teach, but as they used pictures to explain, it was easier for me. Also, the things that I could not ask about, I could understand them from the pictures”*.(professional carer)

Facilitators reiterated training content and key messages to carers and young people during the monitoring visits, which carers found useful and motivating. Carers reported that the young people looked forward to the visits. Carers particularly appreciated that most of the facilitators were experienced in working with people who have intellectual impairments. A professional carer said that facilitators “*should like to work with our children, they should not be afraid of the children*”.

#### 3.1.2. Appropriateness

The appropriateness of the intervention was assessed through expressed intent to continue use and perceived fit within organisational context.

All carers of young people using the campaign components said that they would continue using the resources after the end of the programme.

*“I think she will use [them]. They also think that these are very useful, they feel they are safe after using these”*.(professional carer)

Facilitators from DSSN also expressed a desire to continue implementing the campaign, and suggested ways to take it forward within their existing networks, day care centres, and schools. This is because facilitators felt that the Bishesta campaign filled an existing gap in their support to carers of people with intellectual impairments:

*“…one of the biggest problems, when they have menstruation, some of the families they tie-up the girls at home, because the girl is running here and there. At that time, we didn’t have any ideas of these type of trainings”*.(DSSN facilitator)

Key informant interviews with WaterAid revealed that the Bishesta campaign “add[s] new knowledge” to the WASH sector in Nepal and globally. Interviewees reported that their engagement in the research and campaign had been “inspiring”, and that as WaterAid focuses on MHM and inclusion, the campaign should be continued.

### 3.2. Demand

Demand refers to the extent to which the campaign components are being used by the target group, their perceived level of impact in achieving the target behaviours, and an assessment on potential demand for the intervention.

#### 3.2.1. Actual Use of the Campaign Components

During observation, young people demonstrated an understanding of the campaign components’ purpose and how they should be used. Many felt a great sense of pride and ownership over the materials.

The branded menstrual storage bag was consistently used by carers and young people. All the young people enjoyed using the shoulder bag, which many viewed as a fashion accessory. Carers said that the shoulder bag meant that they were more likely to leave the home with the young person when they are menstruating.

*“I think the [shoulder] bag was most useful for me, as if we need to go out, we don’t have to carry the materials for the girls, they can carry them themselves in the bag. We didn’t use[d] to carry the plastic [bag]. So, carrying the plastic is one of the things that I learnt from the training”*.(professional carer)

The menstrual bin was well understood and used by all participants. It proved critical for improving the hygienic disposal of menstrual pads by young people and carers. The pain bangle was too complicated for the young people to understand. Only one young person was able to differentiate the colours on the pain bangle or understand how they relate to levels of pain. Additionally, some carers were unable to recall what the colours symbolised.

The visual stories and the Bishesta dolls were useful for improving communication on MHM and adopting the target behaviours. ‘I Manage’ visual story was cited a number of times by carers as a positive trigger for young people to use a menstrual product effectively (this relates to target behaviours: Use a menstrual product; do not to show blood in public). Carers regularly looked at the visual stories with the young people between the group training sessions. The visual stories were intended to be used to empower the young person, rather than as a teaching aid. For instance, the young person should hold the book and turn the pages themselves; carers should facilitate communication about the pictures but the young person should come up with their own ideas about what is happening. However, many carers used the books to direct the young person to manage their menstruation more hygienically. Some carers were concerned that the young people with severe intellectual impairments would rip or tear the books, so kept the books in a cupboard rather than next to the young person’s bed.

The majority of carers found the menstrual calendar overly complicated and confusing. However, many were using it to track the start date of the young person’s menstruation. The large and small Bishesta dolls were helpful in teaching the young people practical MHM skills and encouraging them to adopt the target behaviours. Outside the group training, a number of carers practiced changing the small Bishesta doll’s pad with the young person, and some young people did this independently. When observing the young persons’ reaction to the Bishesta doll, most demonstrated that they related to it. For instance, one young person pointed at the doll and then at herself during the observation. Others gestured that the doll is the same character as Bishesta in the visual stories; one person pointed at the campaign logo on the branded menstrual storage bag and then to the Bishesta doll, which the researchers assumed indicated brand awareness.

Carers were asked to rank the campaign components according to which they perceived to have the biggest and least impact on the young person’s target behaviours (Table 3).

All facilitators identified the large Bishesta doll as the most impactful campaign component used during the group training sessions; this was followed by the branded menstrual storage bag. All facilitators graded the pain bangle as the least useful.

#### 3.2.2. Perceived Demand

Three young people who live in the residential institution were not included in the intervention. After the group training sessions, the professional carers independently went through the training content with these young people, using the campaign materials. As the carers were not instructed to do this, it indicates that the campaign components and the way the training was delivered was appropriate, accessible, and demanded by carers.

All professional carers requested that all the young people in their care, and those who required more support to adopt the target behaviours, are involved in any future intervention. Family members asked for the same. These findings suggest that the carers see value in ensuring all the young people can manage their menstruation as independently as possible.

### 3.3. Implementation

Implementation refers to the degree to which the intervention can be delivered as intended within the context [17].

#### 3.3.1. Delivered as Intended

Analysis of the process monitoring data shows that two of the three group training sessions were delivered by the correct number of facilitators (see results for indicator number 5 in Appendix A). The third was delivered by three facilitators instead of five. All other aspects of the campaign, such as all materials, were distributed, all elements of the sessions took place and were delivered as intended. All monitoring visits were carried out, but the facilitators did not visit participants in pairs due to competing work demands outside the campaign.

Carers were satisfied with one group training session per month and said that the length of time for the group training sessions was adequate. Many asked that the campaign could last longer than three months, so that the information could be repeated. Facilitators echoed this.

#### 3.3.2. Amount and Type of Resources Needed to Implement

Working with people who have an intellectual impairment is resource intensive, because simple and clear information must be provided repetitively and people may require one-to-one support from facilitators.

In the Bishesta campaign, the number of facilitators required per young person was considered. For example, in the group training sessions, one facilitator worked with two young people at the same time: One of which had a severe intellectual impairment. This was not effective, as this person required one-to-one support.

### 3.4. Practicality

Practicality explores the extent to which the intervention can be carried out using existing resources [17].

#### Positive and Negative Effects on the Target Audience

(a) Key behaviours: Young person: Uses a menstrual product; Carers: Provide enough menstrual products

Over the short timeframe, two young people wore a menstrual product when they were not previously (see Appendix A for the structured questionnaire results and analyses of the findings against the qualitative data). Qualitative data show that two participants with severe intellectual impairments had an increased understanding about the need to wear a menstrual product, though still required support to place them correctly. However, one young person with a severe intellectual impairment did not use a menstrual product. It is likely that a longer intervention is required to achieve total behaviour change within this sub-group.

Carers reported that the young people felt a greater level of comfort when menstruating and were willing to use a menstrual product because they understood why it was needed. Wearing a menstrual product also made a difference to how the carers perceive the young person. After one young person returned home for a festival, one professional carer explained how parents’ view of their daughter had changed because she managed her menstruation with greater independence, and did not show her blood in public.

*“Her parents told me that she did not do anything bad, they felt that she behaved like a grown up when she was taking care of the menstrual materials”*.(professional carer)

Another carer, who before the intervention, used to restrict the young person’s movements during menstruation because she was worried she would show her menstrual blood in public, took her on the bus with her because she was wearing a menstrual product.

*“I didn’t use to take her when I went somewhere. Because if she menstruates when we are out somewhere, like in a bus, she didn’t put on the pads, then I would be in trouble. Now, we have dustbin, storage bag, shoulder bag. There have been many changes”*.(family carer)

Carers reported a reduction in menstrual hygiene-related caring duties. For instance, there were less clothes and bedding to wash when the young person used a menstrual product. Carers reported that some young people knew where to get a clean menstrual product, and more independently fetched their own menstrual product from the storage bag, washed the used one and dried it in direct sunlight.

*“She learnt many things, she knows where to keep her pads. When I tell her to change her pads and clothes, she does them by herself, and also washes them. She didn’t use the menstrual products before, but after the three trainings, there has been good changes in her behaviour”*.(family carer)

Carers reported preparing for the young person’s menstruation by filling up the menstrual storage bags with clean menstrual products and underwear. In the past, young people either asked carers for a menstrual product, relied on carers to fetch a clean product, or did not use one. An unexpected outcome was carers’ increased understanding of good menstrual hygienic practices.

*“Before the training, though we used to dry [the menstrual product] in sunlight, we didn’t let them dry properly, sometimes we used to dry them inside as well. We didn’t care about killing the bacteria in the sun, we only cared about drying, sometimes we even used to give them the [menstrual] cloths even when it was not fully dried”*.(professional carer)

(b) Key behaviours: Young person: Do not show blood in public

There was a marked improvement in the target behaviour of not showing blood in public. Carers explained that they now dispose of the used menstrual product in the menstrual bin instead of putting it in a plastic bag and throwing it in the open. Young people were also disposing of their used menstrual products in the menstrual bin.

*“Sometimes they used to throw it behind the gate, sometimes they threw it from the hostel window. When we circled the building perimeter, we used to find many used pads. Now, they don’t do that. They know how to dispose”*.(professional carer)

The reason for focusing on this target behaviour was because some people with intellectual impairments in the formative research sample were abused by family members and the general public for showing their menstrual blood in public. This included walking around with blood on their clothes and removing their menstrual product and showing it to others. However, focusing on this behaviour in the Nepali context where menstrual blood is viewed as dirty, contaminating, and shameful, may have unintentionally reinforced these beliefs. Only one carer noted:

*“The notion of shame is not seen much in our children. So, we tell them that they have to have some shame”*.(professional carer)

(c) Key behaviours: Young person: Use pain relief; Carers: Provide pain relief; show love

According to the structured survey, 40% (*n* = 4) more young people always used pain relief after the intervention. The in-depth interviews revealed that young people have a greater understanding that menstruation can cause discomfort, which can be managed with pain relief options. One carer reported that the young person she looks after is now calmer during menstruation.

*“She is upset sometimes when she is feeling pain, and I give her hot water drink and console her and tell her to rest. I don’t let her sleep alone; I sleep with her”*.(family carer)

Carers reported having an increased understanding of pre-menstrual and menstrual symptoms, which has led them to respond more compassionately when the young person is distressed during menstruation.

*“Before, I used to get irritated when they got angry or upset during menstruation. I used to scold them, I thought they are not obeying me and creating problems for me...... But now after the training, I have realized that this is natural, getting angry and upset is natural”*.(professional carer)

(d) Cost analysis

As this is the first intervention of its kind in Nepal and globally, it is difficult to benchmark the campaign costs. Table 4. presents the cost of the pilot study compared to the estimated cost of delivering the campaign at scale. The scale-up costs are based on 25 facilitators delivering 100 group training and monitoring visits to 1000 young people and 1000 carers.

It is estimated that the cost per young person will reduce from $1240–$90 if the campaign is delivered at scale. This could be achieved by (1) economies of scale, (2) a reduction in production salary costs, and (3) one off production costs for training materials, which can be reused across all group training sessions. The facilitators’ salaries reflect the support required by the young people.

During the in-depth interviews, carers were asked if they would be willing to pay for any campaign components themselves, and if so, which? The rationale for this question was to explore how the campaign costs could be reduced. Some carers indicated that they would be willing to pay for the following campaign components: Menstrual pads (38% *n* = 3), shoulder bag (38% *n* = 3), menstrual bin (25% *n* = 2), branded menstrual storage bag (25% *n* = 2), and visual stories (13% *n* = 1). One carer recommended that carers buy the campaign components, and that they are given the Bishesta campaign logo so they can sew it onto the menstrual shoulder and storage bags they purchase. These suggestions were not taken into account in the at scale costs (Table 4).

## 4. Discussion

Findings demonstrate that the Bishesta campaign is acceptable and feasible. For instance, facilitators were satisfied with the training they received on how to deliver the intervention and the campaign manual, which guided their facilitation. This support helped the facilitators feel confident to deliver the campaign as it was intended. Carers and young people valued the visual and practical training methods, materials, campaign components and monitoring visits. MHM information was accessible for the young people and carers with limited literacy levels. Facilitators from the DSSN expressed a desire to continue delivering the Bishesta campaign, as they reported that it filled an existing gap in their self-care packages for people with intellectual impairments and carers. WaterAid staff also felt that the campaign could contribute positively to MHM knowledge in Nepal and globally.

Findings indicate a demand for the Bishesta campaign. Participants used 80% of campaign components; these components made the target behaviours attractive and enabled participants to carry them out with ease. The large Bishesta doll and the visual stories were identified as the most impactful campaign components for changing young people’s target behaviours.

The campaign was largely implemented as planned. The number of facilitators to young people during the group training sessions worked well, as one-to-one support could be provided when required.

In terms of practicality, there were improvements across all target behaviours and participants were able to carry out the intervention activities. A possible negative effect on the target groups was the focus on ‘do not show blood in public’ target behaviour, as this may have inadvertently reinforced cultural beliefs that menstrual blood is dirty and shameful. The cost of producing the campaign components and training materials is arguably acceptable, although a benchmark is not available.

The focus of this study was feasibility, not limited-efficacy [17]. However, indicative outcomes from this small sample are that young people’s levels of confidence, comfort, and autonomy during menstruation may have increased, which arguably leads to greater agency; carers’ menstrual-related caring duties have reduced, meaning they feel less overwhelmed. Carers reported a greater level of understanding of the young person’s pre-menstrual and menstrual symptoms, which meant they responded compassionately when the young person experienced these. Carers also expressed an ability to provide a higher quality of menstrual care as a result of the campaign.

A recent systematic review conducted to assess of MHM requirements of people with disabilities, and the barriers they face, revealed that (1) people with intellectual impairments face particular barriers to MHM and (2) limited MHM interventions exist for this group [14]. Specific barriers highlighted through the review is a dearth of MHM information, training, and support for people with intellectual impairments and their carers, limited carer’s understanding of severity of pre-menstrual symptoms experienced by people with intellectual impairments and few MHM interventions for this group. The Bishesta campaign contributes to addressing these gaps by providing accessible and practical MHM information, training, and support for people with intellectual impairments and their carers in Nepal, with a focus on improving carers’ understanding of pre-menstrual and menstrual symptoms experienced by the young people, and how to manage these.

### 4.1. Strengths and Limitations

A key strength of this research is that the development of the Bishesta campaign followed the behaviour-centred design model, so it is systematic, evidence based, and relevant for the context. The majority of the young people (80%, *n* = 8) and carers (75%, *n* = 6) were involved in the formative research, the intervention, and the feasibility study, so a strong level of trust and rapport was established between them and the lead author. This may have led carers to be overly positive in the evaluation, but the in-depth interviews with carers were complemented by the quantitative survey and observation of young people, which allays some concerns about researcher bias.

In terms of other limitations, the timeframe was tight. This meant that the data collection was conducted shortly after the completion of the last group training session, when the campaign’s key messages were fresh in people’s minds. Therefore, the findings should be viewed as indicators of behaviour change, rather than actual behaviour change. The sample size was small, so findings are not generalisable. The lead author led all aspects of the study, including developing the intervention and evaluating its feasibility and acceptability. Concerns of researcher bias were managed by close supervision provided by independent academic staff at the London School of Hygiene and Tropical Medicine, and triangulation of qualitative findings with quantitative data, as discussed above. Facilitators were interviewed as part of the feasibility and acceptability study. Their responses to questions may have been influenced by their desire for continued involvement in the intervention if it is scaled up. The lead author explained that the intervention might not be delivered further, but that if it is, the facilitators may not be involved in the scale up. This was reiterated throughout the key informant interviews.

### 4.2. Implications for Further Research

This study focused on assessing the Bishesta campaign against the first four of eight areas of focus in Bowen et al.’s feasibility study framework [17]. This section sets out the implications for further research against the remaining four areas.

(a)Adaptability: A number of changes should be made to the intervention: (1) The menstrual calendar should be simplified, (2) the pain bangle should be excluded, (3) the training provided to facilitators should include a greater emphasis on using positive language about menstruation, and (4) the cost of producing the reusable menstrual pads should be reduced by comparing the prices and quality of goods across a number of providers and selecting a more affordable option.(b)Integration: Findings from this study indicate the importance of facilitators understanding the lived experiences of the young people and carers, beyond MHM. Within this area of focus, facilitators’ skill set and previous experience of working with the target groups should be tested. Disability service providers could deliver the intervention within their wider self-care programmes, aimed at enabling people with intellectual impairments to live as independently as possible. This could be supported by WASH sector actors to ensure the MHM content is in line with wider efforts to promote effective MHM, including challenging harmful traditional beliefs, practices, and menstrual taboos, as well as ensuring access to WASH services to support menstrual hygiene management practices. The process of delivering the intervention should be monitored to assesses the level of organisational system change (e.g., organisational structure, strategies, operational methods, technologies) required to integrate the campaign into existing organisational commitments.(c)Expansion: The Bishesta campaign was tested with a small sample size. Now it should be delivered at a wider scale with people who have an intellectual impairment and their carers living in different locations in Nepal.(d)Limited-efficacy: An impact evaluation is needed to understand if the Bishesta campaign leads to sustained behaviour change. This could be designed as pre-post design study. A mixed methods baseline survey to understand current MHM behaviours related to the target behaviours should be conducted before the intervention. The campaign should be delivered and then evaluated using quantitative and qualitative methods to understand behaviour change outcomes.

## 5. Conclusions

The Bishesta campaign presents an opportunity to ensure that one of the groups most vulnerable to exclusion from MHM interventions is not left behind. The campaign begins to fill the documented gap in the sexual and reproductive health provision for people with intellectual impairments, and MHM information and support for their carers. It should now be tested with a larger sample and evaluated to understand its efficacy.

## Figures and Tables

**Table 1 ijerph-16-03750-t001:** Bowen’s feasibility study framework.

Area of Focus	Topics to Investigate	Outcomes of Interest
Acceptability	How the participants and implementers react to the intervention	Satisfaction; Intent to continue use; Perceived appropriateness; Fit within organisational culture; Perceived positive or negative effects on organisation.
Demand	Estimated or actual use of intervention activities in a defined target group	Actual use; Expressed interest or intention to use; Perceived demand.
Implementation	The extent, likelihood, and manner in which an intervention can be fully implemented as planned and proposed	Degree of execution; Success or failure of execution; Amount and type of resources needed to implement; Factors affecting implementation ease of difficulty; Efficiency, speed, or quality of implementation.
Practicality	The extent to which an intervention can be delivered when resources, time, commitment, or a combination of these are constrained in some way	Positive/negative effects on target population; Ability of participants to carry out intervention activities; Cost analysis.

**Table 2 ijerph-16-03750-t002:** Household monitoring visit indicators.

Number	Indicator	Yes/No	Method
1	Shoulder bag near the young person’s bed		Observe
2	Young person can identify the shoulder bag		Ask young person
3	Branded menstrual storage bag near young person’s bed		Observe
4	Young person can identify where to get clean menstrual products		Ask young person
5	Branded menstrual storage bag is well stocked		Observe
6	Shoulder bag well stocked with menstrual products		Observe
7	Carers using menstrual calendar to track menstruation		Observe
8	This household has Yes against questions 1–7		
9	This household has Yes against questions 1, 3, 5–7		

**Table 3 ijerph-16-03750-t003:** Perceived levels of impact.

Variable	Component	Number of Carers
Most impactful	Large Bishesta doll	6
	Visual stories	2
	Shoulder bag	1
Least impactful	Pain bangle	8
	Shoulder bag	1

**Table 4 ijerph-16-03750-t004:** Bishesta campaign pilot study costs versus at scale costs (USD).

Activity	Detail	Total Cost: Pilot Study (*n* = 10 Young People)	Total Cost: Scale Up (*n* = 1000 Young People)
Production	Campaign components	$852	$40,410
Production	Training materials	$111	$2103
Salaries	Production team	$2100	$900
Salaries	Facilitators	$5034	$25,169
Training	Facilitators	$231	$750
Delivery	Group training and monitoring visits	$4068	$20,342
Grand total		$12,396	$89,674
Cost per young person		$1240	$90

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
