# Peer review of "Feasibility Study of a Menstrual Hygiene Management Intervention for People with Intellectual Impairments and Their Carers in Nepal"

_ijerph, 2019, doi:10.3390/ijerph16193750_

Round 1

Reviewer 1 Report

This paper describes part of a feasibility study for an intervention aiming at improving menstrual management and hygiene in Nepalese women with intellectual impairments and there caregivers. The topic is relevant and the paper is well written. Below are my suggestions regarding major and minor issues that could be improved.

Major issues

I may have missed something regarding political correct language, but I wonder why you call the young women participating in your study “people” and not “women”. People is a gender-neutral term but this is clearly not an issue affecting both genders. While you mention some particular barriers affecting young women with intellectual impairments in the discussion, I would suggest to present those in the introduction already (line 72). How were study participants recruited? In Table 1, you introduce indicators for household monitoring but results from that part are never presented. Please expand on data analysis and interpretation. What do you understand by thematic analysis, did independent coders analyze the transcripts, how were the data on observations of the young women analyzed? I do not understand lines 261-264. Three young women were not initially included but later their caregivers went through the materials with them. How does this indicate that the training was accessible, appropriate and demanded? It may not hold true for many other young women in need. Lines 274-276: Please explain why five facilitators were enough but three not and how you found out. Sometimes you differentiate between mild, moderate and severe intellectual impairment. How did you define and determine this?

Minor issues

Line 28, delete size. Lines 102-103: You don’t need both aim and focus of the article. Please also specifically mention the four components of feasibility you investigated. Line 112: four, not for. Line 112-113: Here and at other places something seems wrong with references. Line 252: then, not the.

Reviewer 2 Report

It is very difficult to assess behavior changes of the people within a short period of time, specifically for the people with disability. Moreover, the reliability is questioned when one single-use was observed to measure the behavior change of the disabled person. In that case, a good quantitative analysis is required for the carers, facilitators and, logistics delivered for the pilot study.

To judge the reliability, carers attitudes towards the disabled is important rather than the behavior of individuals with disabilities.     

A few additions:

1. Attitude analysis of the carers towards the disabled people and their relation with the disabled people.
2. Detail information on carers ( their level of education and other demographics)
3. Revision by a professional English editor for improvement of English and spelling check ( for instance, 112 number of the line: there will be "four" instead of "for")

Round 2

Reviewer 1 Report

The authors have well responded to the reviewer’s comments. Only a few minor revisions remain:

Line 28/abstract: Within the sample, not within the sample size Lines 86ff: It is suggested to move the paragraph on participants and their recruitment to the respective section I Methods, i.e. 2.1.2. Line 112: Appropriateness, not appropriate. Lines 493-495: I don’t understand how demographic information on the study population can be used to determine if they are representative of the target population if this information is not available for the target population. The only way to ensure representativeness is drawing a random sample from the target population. S2: Use endpoint instead of endline.

Author Response

Thank you very much for reviewing the article for the second time. Below are my responses to each of your comments:

Point 1: Line 28/abstract: Within the sample, not within the sample size

Response: Thank you for highlighting this; ‘size’ has been deleted from line 28.

Point 2: Lines 86ff: It is suggested to move the paragraph on participants and their recruitment to the respective section I Methods, i.e. 2.1.2.

Response: Thank you for your suggestion. Details of how the participants were recruited is now detailed in lines 125-135.

Point 3: Line 112: Appropriateness, not appropriate.

Response: Thank you. 3.1.2 has been updated to ‘appropriateness’ instead of appropriate.

Point 4: Lines 493-495: I don’t understand how demographic information on the study population can be used to determine if they are representative of the target population if this information is not available for the target population. The only way to ensure representativeness is drawing a random sample from the target population.

Response: Thank you for highlighting this. I have deleted lines 493-495.

Point 2: S2: Use endpoint instead of endline.

Response: ‘Endline’ has been changed to ‘endpoint’ in S2.

Reviewer 2 Report

This article has improved than the previous version. Authors are adequately answered the raising points. 

Author Response

Thank you very much for reviewing the article and second time and for your final comments.